# Experimental and Bioinformatic Insights into the Effects of Epileptogenic Variants on the Function and Trafficking of the GABA Transporter GAT-1

**DOI:** 10.3390/ijms24020955

**Published:** 2023-01-04

**Authors:** Dolores Piniella, Ania Canseco, Silvia Vidal, Clara Xiol, Aránzazu Díaz de Bustamante, Itxaso Martí-Carrera, Judith Armstrong, Ugo Bastolla, Francisco Zafra

**Affiliations:** 1Centro de Biología Molecular Severo Ochoa, Facultad de Ciencias, Consejo Superior de Investigaciones Científicas, Universidad Autónoma de Madrid, 28049 Madrid, Spain; 2IdiPAZ, Institute of Health Carlos III (ISCIII), 28046 Madrid, Spain; 3Sant Joan de Déu Research Foundation, 08950 Barcelona, Spain; 4IPER, Institut de Recerca Pediàtrica Hospital Sant Joan de Déu, 08950 Barcelona, Spain; 5Genetics Unit, Hospital Universitario de Móstoles, 28935 Madrid, Spain; 6Pediatric Neurology, Hospital Universitario Donostia, Biodonostia, Universidad del País Vasco UPV-EHU, 20014 San Sebastián, Spain; 7Molecular and Genetics Medicine Section, Hospital Sant Joan de Déu, 08950 Barcelona, Spain; 8CIBER-ER (Biomedical Network Research Center for Rare Diseases), Institute of Health Carlos III (ISCIII), 28029 Madrid, Spain

**Keywords:** GABA transporter, epilepsy, intracellular trafficking, 4-phenylbutyrate

## Abstract

In this article, we identified a novel epileptogenic variant (G307R) of the gene *SLC6A1*, which encodes the GABA transporter GAT-1. Our main goal was to investigate the pathogenic mechanisms of this variant, located near the neurotransmitter permeation pathway, and compare it with other variants located either in the permeation pathway or close to the lipid bilayer. The mutants G307R and A334P, close to the gates of the transporter, could be glycosylated with variable efficiency and reached the membrane, albeit inactive. Mutants located in the center of the permeation pathway (G297R) or close to the lipid bilayer (A128V, G550R) were retained in the endoplasmic reticulum. Applying an Elastic Network Model, to these and to other previously characterized variants, we found that G307R and A334P significantly perturb the structure and dynamics of the intracellular gate, which can explain their reduced activity, while for A228V and G362R, the reduced translocation to the membrane quantitatively accounts for the reduced activity. The addition of a chemical chaperone (4-phenylbutyric acid, PBA), which improves protein folding, increased the activity of GAT-1WT, as well as most of the assayed variants, including G307R, suggesting that PBA might also assist the conformational changes occurring during the alternative access transport cycle.

## 1. Introduction

GABA is the major inhibitory neurotransmitter in the brain, and dysfunction of GABAergic signaling results in diverse forms of epilepsy, autism and intellectual disability [1,2,3]. GABAergic signaling is terminated by GABA clearance from the synaptic cleft by several transporters (GAT-1, GAT2 and GAT3), with GAT-1 the most prominent in this function. GAT-1 localizes in GABAergic axons and nerve terminals, locations achieved by mechanisms that control intracellular trafficking and the lateral mobility of the transporter within the neuronal membrane [4,5,6,7,8]. GAT-1 has also been detected in astroglial processes, in oligodendrocytes and in microglia [9].

The 3D structure of mammalian GAT-1 bound to the specific inhibitor tiagabine was recently determined by single-particle cryo-EM [10]. These data indicate that GAT-1 folds in a manner similar to the bacterial ortholog LeuT, as well as to the eukaryotic transporters for dopamine (from *Drosophila melanogaster*) [11], serotonin or glycine (from humans) [12,13]. Indeed, this folding is widely distributed in nature, not only in the SLC6 family of eukaryotic transporters, but also in the SLC5 or SLC7 families, as well as in several prokaryotic transporters [14]. Briefly, 12 transmembrane (TM) helices are organized in two halves, showing an inverted pseudo-twofold symmetry, with the symmetry axis lying parallel to the membrane plane running through the center of the transporter. Four helices (TM1, 2, 6 and 7) form a bundle (referred to as the core bundle) that is tilted with respect to the rest of the protein. The six remaining helices in the core of the protein (TM3−5 and TM8−10) form a scaffold that wraps around three sides of the core bundle [15,16]. Substrates and sodium ions are coordinated between the scaffold and the core domain. Specifically, helices 1 and 6 are partially unwound, defining TM1a-TM1b and TM6a-TM6b, respectively, which favors the formation of a hinge that contains part of the binding site for the substrate and cotransported Na^+^ [17]. The GAT-1 structure indicates that the uncoiled segments of TM1 include residues C58 to G63, while the unwound segment linking TM6a and TM6b corresponds to residues G297 to G301 [10]. The transport cycle proceeds through an alternating access mechanism, in which the transporter exposes its primary binding site on each side of the membrane. The outward-open conformation, contributed by TM1b, TM6a and TM7, permits the binding of substrates and ions from the extracellular compartment. Subsequently, the extracellular gate closes (occluded conformation), and finally the intracellular gate opens (inward-open conformation), releasing substrates and ions into the cytoplasm. Binding of tiagabine stabilizes the protein structure in the inward-open conformation (PDB: 7sk2) [10].

GAT-1 plays a prominent role throughout development in controlling neuronal excitability and brain connectivity. GAT-1 is encoded by the *SLC6A1* gene, and studies performed in recent years have identified several dozen variants of this gene, some of which are associated with autism or diverse epilepsy syndromes, with myoclonic atonic epilepsy (MAE) and intellectual disability as prominent features [1,2,18]. The function of some of these mutants was shown to be decreased in GABA uptake assays [19]. However, the pathogenic mechanisms linking these deficiencies with the transporter structure remained unknown until recently, when several reports were published. First, the variant G236S (impaired transporter stability, resulting in reduced cell-surface and total GAT-1 levels) was associated with Lennox–Gastaut syndrome, a severe form of epilepsy with developmental delay [20]. A second study showed that the variant P361T destabilizes GAT-1, which was retained in the endoplasmic reticulum (ER) and probably degraded [21]. Finally, three reports appeared during the preparation of the present work, which analyzes five additional missense variants (V125M, S295L, G362A, D410E and L460R) and two truncated ones (E16X and W495X) [22,23,24]. Importantly, most of the missense mutants were partially activated by the chemical chaperone 4-phenylbutyrate (PBA) under different conditions that mimic the homozygous and heterozygous expression of the mutant [22]. PBA is known to prevent the misfolding of several proteins, including the transporters for glycine, serotonin and creatine [25,26,27,28].

As a result of the diagnostic activity of our hospital centers, we found two patients suffering from MAE and intellectual disability who were carriers of a new variant of the *SLC6A1* gene (G307R). In this study, our first objective was to characterize the pathogenic mechanisms of this variant, as well as other variants of this gene that are also poorly characterized and have similar pathological consequences. By using heterologous expression systems (cell lines and neurons in culture), we have studied the relationship between the position of the different mutations in the structure of the GABA transporter and its effect on the activity and intracellular trafficking of these variants. In addition, we applied bioinformatic methods to try to correlate the positions of the mutations with the dynamics of the functionally important sites of the protein: the extracellular and intracellular gates and the neurotransmitter binding site. Finally, given the therapeutic potential of PBA, another objective of this study was to analyze the effect of this compound on the activity of the different mutants studied.

## 2. Results

### 2.1. Patients with the Variant G307R of the SLC6A1 Gene

Two independent clinical studies among a cohort of patients with autism associated with epilepsy identified two unrelated patients with a new variant of the GABA transporter GAT-1, encoded by the *SLC6A1* gene.

The first patient was a 47-year-old Spanish Caucasian woman, born after normal pregnancy, delivery at term and eutocic. The body weight at birth was 3200 g and was observed in the Department of Pediatric Neurology due to global delay, with favorable evolution. Before six months of age, she presented global hypotonia. The first signs of global delay appeared around six months of age, and the clinical signs were initially interpreted as ataxia-spastic cerebral palsy. She started walking at two years old, with instability, with support and not autonomously. By five years old, she showed global hypotonia, ataxia and distal tremor of the hands and lack of balance backwards, as well as convergent strabismus. Soon after, at five and a half years old, epilepsy debuted, with sleep difficulties and absence seizures. Phenobarbital and benzodiazepine were used to reduce to one to two seizure episodes per week. After a year, the treatment changed to ethosuximide because she had six to eight seizures per day. There were difficulties in controlling absence seizures by treatment with valproate (VPA) plus ethosuximide. The electroencephalogram (EEG) obtained at seven years of age indicated generalized spike and wave discharges (SWDs) at 3 Hz, lasting 3–4 s. She attended a special school, where she was sociable until she was 14–15 years old, at which point she presented a sudden change of character, with a tendency to self-harm, which improved with Zyprexa (olanzapine). At 24 years old, seizures were controlled only with VPA, and magnetic resonance imaging was normal. When she was 31, diabetes and hypothyroidism appeared and were treated with Metformin and Eutirox (levothyroxine). Given the persistence of a global delay of unknown etiology that could be attributed to an atypical form of Rett Syndrome, at 36 years of age, a study of the *MECP2*, *CDKL5*, *ARX* and *NTNG1* genes was carried out, with negative results. In addition, no alterations were found for this patient in the comparative genome hybridization (CGH) array (400K) https://bioarray.es/es/info/microarrays-de-adn-48) (accessed on 20 November 2022). She was later included in a whole-exome sequencing (WES) study, identifying a *SLC6A1* variant (NM_003042.3: c.919G>A) leading to an amino acid change (NP_003033.3: p.Gly307Arg). At the time of sequencing, the c.919G>A variant was absent in the Genome Aggregation Database (gnomAD; gnomad.broadinstitute.org). This variant was classified in ClinVar as a likely pathogenic variant, but the possibility that it may be a rare benign variant could not be excluded. Based on the pathogenicity classification of the American College of Medical Genetics and Genomics and the Association for Molecular Pathology (ACMG/AMP) guidelines, this variant was predicted to be likely pathogenic (PP2, PP3, PP5, and PM2). A trio WES study and Sanger sequencing of parental samples indicated that c.919G>A occurred de novo in the patient (Figure 1).

Patient 2 was a 2.5-year-old Spanish Caucasian boy with no family history of neurodevelopmental disorders. He was born by cesarean section at 37 weeks of gestation due to vasa previa after a normal pregnancy, with a birth weight of 2860 g. He was referred to the clinic at 9 months old due to psychomotor delay with generalized hypotonia and poor social contact. He also had sleep disturbances and refused solid food. Neurologic examination revealed no dysmorphic features, poor visual contact, impaired social interaction and midline stereotypies, such as handwashing or hand interlocking. He had a generalized hypotonia, including inability to sit independently and absence of the Landau reflex.

At 15 months of age, daily episodes of ocular elevation and possible disconnection were observed. Brain magnetic resonance imaging did not show abnormalities. EEG was abnormal due to a slower background rhythm and frequent occipital bursts of delta waves along the entire tracing, with interspersed occipital spike-wave activity. The episodes disappeared after VPA treatment, although the slow occipital delta activity persisted on the EEG. At 22 months, clobazam was added due to the appearance of palpebral myoclonus and 3–4-min lasting absences. Thereafter, he had no new paroxysmal episodes.

Currently, at the age of 4, he is at a motor level where he can walk independently, though ASD features persist, with behavioral disorder, motor stereotypes and no language.

Consequently, a diagnostic exome sequencing for epileptic encephalopathies (Bioarray, 543 genes) was requested. Results revealed the same mutation in gene *SLC6A1* as in patient 1 (NM_003042.3: c.919G>A; NP_003033.3: p.Gly307Arg). A trio-based WES also revealed that it was a de novo variant. The main clinical characteristics of the two patients are summarized in Table 1.

While this work was in progress, the same variant was found in another patient in a study describing 21 novel variants of this gene, most of them concentrated in TM6 and TM7 [29].

### 2.2. Effect of Various SLC6A1 Mutations on Transporter Function and Trafficking

Residue G307 is located in the intracellular part of TM6b and, therefore, close to the residues involved in the opening and closing of the intracellular gate (Figure 2A). We decided to study the structure-function relationships of this variant, firstly, by comparing it with other variants, which are located in the vicinity of the permeation pathway (TM6-7) and are equally poorly characterized from the point of view of their structure-function relationships. In this series, we investigated variants A288V, G297R (both in TM6) and A334P (in TM7), apart from G307R (red residues in Figure 2A).

Subsequently, we compared them with other epileptogenic variants also located in the inner half of the transporter, relatively far from the permeation pathway (variants A128V (TM3), G232V (TM4-IC2) and G550R (TM12)) (blue residues in Figure 2A,B). Transporter activity was tested in HEK293 cells overexpressing each mutant, and a dramatic drop (below 5% of WT) was found in the activity of G297R, G307R and A334P; in this group, A288V was the most active (~25% of WT uptake). The activity of the mutants of the second group was also severely reduced, especially for the A128V and G550R mutants, with G232V being somewhat better (~15% of WT) (Figure 2C).

To investigate the cause of the altered function of the various mutants, we performed western blot on cellular lysates obtained from HEK293 cells transiently transfected with the aforementioned mutants (Figure 3A). In all cases, we observed a band at ~46 kDa, an immature form of the transporter, which, according to previous studies [30], corresponds to the core glycosylated protein (after the dolichol-linked precursor oligosaccharide is transferred to each of the three N-glycosylation sites of GAT-1 that are used in the ER). In the first group, all mutants, except for G297R, presented variable amounts of a second smeared band between 55 and 75 kDa that corresponds to completely glycosylated forms of GAT-1. In the second group, A128V and G550R presented only the immature band, whereas G232V displayed both. Quantification of both the mature and the immature protein is presented in Figure 3B. Relative to the mature bands, G307R was indistinguishable from WT, while A334P and G232V both presented significant amounts of the glycosylated band. The rest of the variants presented a very low amount of mature protein (less than 10% of WT). However, the level of immature bands was quite constant in all the variants, except for G297R, which seemed to be unstable.

To confirm that the mature protein indeed corresponded to transporter molecules located on the cell surface, biotinylation assays were performed using the membrane-impermeable reagent sulfo-NHS-biotin. Biotinylated protein, as well as samples of the cellular lysate and the non-biotinylated fraction (flow-through), were analyzed by western blot (Figure 3C,D). In these experiments, besides monomeric forms of the transporter, aggregates of the mature and immature GAT-1 were also observed, although their quantities varied from experiment to experiment, suggesting that aggregation occurred artifactually during the handling of the samples for biotinylation. In any case, in terms of trafficking to the cell surface, aggregates behaved like the monomeric forms: aggregates of immature forms were not biotinylated efficiently, while those of mature forms were labelled. Consistent with the western blot, mutants A128V, G297R and G550R did not reach the membrane, and the protein was mostly recovered in the flow-through of the agarose-streptavidin resin (Figure 3C). Nevertheless, small amounts of the core-glycosylated form were detected on the membrane for A128V and G550R (Figure 3D). This was also observed for A288V, which, moreover, presented low levels of glycosylated protein (mainly observed as aggregates). The G307R mutant behaved like the WT, while modest amounts of G232V and A334P reached the cell surface (Figure 3C,D). As a control for correction of the transfection efficiency and to confirm the integrity of intracellular trafficking pathways, each mutant was co-expressed with an unrelated and presumably membrane-reaching transporter, a mCherry-tagged form of the dopamine transporter (mCherry-DAT). The amount of this transporter reaching the cell surface was used as an internal reference to quantify the fraction of GAT-1 variants in the membrane. Therefore, after probing with anti-HA, blots were reprobed for immunoreactivity with anti-mCherry. Two bands were again observed for mCherry-DAT. The lower one (75 kDa) corresponded to the intracellular core glycosylated protein, while the 110-kDa band was the mature form that reached the surface, regardless of which GAT-1 mutant was co-expressed. Aggregate forms of mCherry-DAT were seldom observed. Figure 3E shows the densitometric quantification of biotinylated GAT-1 bands (monomers plus aggregates) relative to mCherry-DAT bands (monomers), confirming that G307R efficiently reached the membrane, whereas the amounts of G232V and A334P were only moderate.

In summary, the biotinylation experiments indicated that the upper band observed by western blot corresponded to a totally glycosylated protein located on the cell surface and was, therefore, responsible for the transporter activity. The variants retained in the ER were not active because they did not meet the substrate. Comparing the activity data (Figure 2C) with the amount of protein on the surface (Figure 3E), it can be seen that among the variants that reached the surface, G307R and A334P did so in an inactive form, while G232V was partially active. Surprisingly, G288V was more active than its biotinylated protein levels indicated.

To confirm these locations by direct visualization, immunofluorescence assays were performed (Figure 4 and Figure 5). COS7 cells were used in these experiments because the small size and large nucleus of HEK293 cells precludes an accurate localization of the immunofluorescence signal. In addition, the location of GAT-1 mutants was compared to that of DsRed2-ER (a fluorescent protein that localizes to the ER) (Figure 4 and Figure 5) or mCherry-DAT, which was again cotransfected along with the different mutants to visualize the plasma membrane (see colocalization in Appendix A).

While GAT-1 WT localized to the cell surface (Figure 4A; Appendix A), mutants A128V, A288V, G297R and G550R were retained in the intracellular compartment, showing a typical pattern of ER staining and colocalization with the marker DsRed2-ER (Figure 4B,D and Figure 5A,D). The immunofluorescence observed for mutant G307R was indistinguishable from that of the WT (compare Figure 5B and Appendix A with Figure 4A and Appendix A, respectively), whereas weaker membrane labeling was observed for A334P (Figure 5C). However, A232V was only seen on the membrane in cells with low GAT-1 expression, whereas it was retained in the intracellular compartment in cells with higher levels of expression, suggesting a poor interaction with the trafficking mechanism. We also analyzed the distribution of the seven GAT-1 mutants in neurons by immunofluorescence (Figure 6 and Figure 7). Again, mutant or GAT-1 WT were cotransfected with mCherry-DAT, which is a control of distribution through the dendritic and axonal compartments. While GAT-1 WT colocalized extensively with mCherry-DAT, being found in cell bodies, dendrites and axonal tracts, mutants A128V, A288V, G297R and G550R were confined to the intracellular compartment of the cell body and major dendrites, and they did not penetrate distal dendrites nor axons. Weak staining of dendrites was occasionally observed for mutants G232V and A334P.

In summary, the G307R variant did not importantly affect its trafficking, but it had a profound impact on its uptake activity. A334P also reached the membrane, although in moderate levels, and was also inactive. Low to moderate amounts of G232V reached the surface, with partial activity. Mutants A128V, A288V, G297R and G550R were highly deleterious for intracellular trafficking, being retained in the ER and completely inactive, except A288V, which showed higher activity than expected.

### 2.3. Bioinformatic STUDIES on the Effect of Mutations in the SLC6A1 Gene

#### 2.3.1. Prediction of Protein Stability Changes upon Mutation

In addition to the seven variants studied here, three recent articles described the pathogenic mechanism of several additional missense mutations of GAT-1 (Table 2). To investigate whether it is possible to predict the effect of these mutations on either trafficking or function, we used a series of bioinformatic tools that calculate the effect of mutations on protein stability and structure (see Methods). These predictions were based on a change of stability between the WT and the mutated protein, assessed as the difference in Gibbs free energy between the native and the unfolded states of the protein (ΔΔG). Some methods, such as Dynamut and Dynamut2, display 3D representations of the predicted atomic interactions for the WT and the mutated form of the protein (see examples of the use of Dynamut2 for position G307 as Appendix A). We chose the convention that scores represent −ΔΔG, so that negative scores predict that the mutation is destabilizing (Table 2). As expected, each tool predicted that almost all of the 13 analyzed mutations destabilize the protein. We examined whether this predicted destabilization could explain the measured fraction of protein that is transported to the membrane. In this case, we would expect that less stable proteins are less present in the membrane, i.e., we would expect a positive correlation between the predicted score and the measured membrane fraction. However, for all but two of the predictors, the correlation was negative, although not significantly, except for two of them (Dynamut and DUET), which showed a weakly significant correlation (−0.49 and −0.47, respectively). These results were related to each other because the predictions of Dynamut are based on a consensus between DUET and EnCOM. As expected, the correlation for MaestroWeb and PopMusic was positive, but not significant (0.28 and 0.17, respectively). We conclude that the predicted changes of stability cannot rationalize the transport to the membrane, either because they are not accurately predicted in the membrane or because other biophysical or biochemical aspects come into play.

Similarly, the transport to the cell surface could not be rationalized through the predictions of the PolyPhen2 program, a method that predicts the phenotypic effect of mutations based on structural and evolutionary criteria, because the mutants G550R or G297R, which presented the greatest retention in the ER, were predicted to be the least harmful (Table 2). As these predictions are qualitative, we could not compute any correlation coefficient.

#### 2.3.2. Analysis of the Structural Effects of Mutations with Normal Modes

Next, we applied linear response theory to the protein native state modeled as an elastic network model in order to predict the structural change caused by the GAT-1 mutations [31]. To this end, we represented the protein through the Torsional Network Model (TNM) [32] and modeled the perturbation induced by the mutation through the combined effect of changes of amino acid size, stability and optimal distance of all the native contacts formed by the mutated residue (*Lorca I*, *Arenas M and Bastolla U 2022. Structure and stability constrained substitution models outperform traditional substitution models used for evolutionary inference. Submitted*). In this way, we predicted two scores: the Root Mean Square Deviation (RMSD) caused by the mutation and the harmonic energy barrier (ΔE) between the native structure of the WT and mutant proteins. Both scores showed a negative correlation with the quantified transport to the membrane, as expected, but they were not significant (r = −0.26 for RMSD and −0.08 for ΔE) (Table 2).

We then used the program TNM to predict the dynamical couplings [33] between the mutated sites and three functional sites of GAT-1: the ligand binding site, characterized as the binding site of the inhibitor tiagabine in the PDB structure with code 7sk2 [10]; the intracellular gate (residues R44, W47, V240, N310, D410); and the extracellular gate (W68, R69, T290, Q291, D451, S456, D457). We considered three types of couplings: co-directionality (the atoms of the two sites tend to move in similar directions, quantified through the Boltzmann average of the scalar product of their direction of motion); coordination (the distance between pairs of atoms show little fluctuation, as expected if they both bind the same molecule; this coupling is positively related with co-directionality); and deformation (predicted structural change on one site induced by a unit perturbation applied to the other site, which is inversely related to both co-directionality and coordination).

In the available conformation of GAT-1 (PDB 7sk2), the tiagabine binding site, which is thought to overlap the GABA binding site, was strongly positively correlated and co-directional with the extracellular gate, and perturbations applied to either site had little influence on the other (Figure 8A). On the contrary, the intracellular gate had zero or negative co-directionality (Figure 8A) and low coordination (Figure 8B) with both the tiagabine binding site and the extracellular gate, and high deformation with the tiagabine site (Figure 8C). This was at first surprising because the conformation that we used to model the native state is inward open. Nevertheless, these couplings are consistent with the two-step model of tiagabine inhibition that has been proposed (see Figure 4 of [10]). The strong deformation coupling between the tiagabine binding site and the intracellular gate is consistent with the hypothesized rearrangement of GAT-1 to the inward-open conformation upon tiagabine binding, and their negative co-directionality and low coordination are consistent with the proposal that tiagabine stalls the transporter in the inward-open conformation, blocking the release of the inhibitor inside the cell.

We then examined the dynamical couplings of the mutated sites examined above, plus three additional GAT-1 mutants (I405V, L415I and G533A) used as negative controls (considered benign in the Gnomad database, though without empirical support).

Of these 16 mutated sites, 10 had both positive co-directionality and high coordination with the tiagabine binding site (Figure 9A,D), underscoring the functional role of the binding site, which tends to be dynamically correlated with many residues, as many functional sites in proteins are [33], and with the extracellular gate (Figure 9B,E). The same sites also had negative or zero co-directionality and low coordination with the intracellular gate (Figure 9C,F).

To consider the effect of the mutations, we segregated them into three classes: pathogenic mutants that were observed at the membrane in very low amounts, such that their detrimental effect is likely to be explained by reduced transport to the membrane (m1: V125M, A128V, G232V, A288V, S295L, G297R, P361T and G550R); pathogenic mutants that were present on the membrane at WT levels (m2: G307R, A334P, G362R, D410E and L460R); and those that are considered benign (m3: I405V, L415I and G533A). Deformation coupling, which tells how much a generic perturbation (mutation) at the target site influences the structure of the functional sites, gives the most relevant information about the effects of the mutation. However, this must be complemented with the predicted global effect of the mutation in terms of RMSD and ΔE, where the specific amino acid change is modeled to compute the structural perturbation applied to the mutated site (Figure 10).

One can see that some mutations are predicted to simultaneously produce a large generic deformation of the intracellular gate and a large global RMSD associated with the specific mutation: G232V, G307R, A334P, I405V and L415I. Nevertheless, three mutants (G232V, I405V and L415I) presented a low ΔE, meaning the native structure of the WT remains dynamically accessible in the mutated protein. In particular, L415I and I405V are conservative mutations associated with a very small ΔE. The two mutations G307R and A334P, which presented large deformation coupling with the intracellular gate and a large predicted RMSD and DE, belong to the group m2 of pathogenic mutants that are transported to the membrane, and the predicted effect of the mutation on the intracellular gate may contribute to explaining why the protein had diminished function.

Therefore, analysis of the native dynamics and the effects of the mutations predicted by the TNM may rationalize the phenotypic effect of the mutations, but experimental verification is still needed.

### 2.4. Rescue of GABA Uptake by the Chemical Chaperone 4-Phenylbutyrate

The chemical chaperone PBA is a potential treatment for epilepsy patients with mutations in the *SLC6A1* gene, and it has previously been shown to increase the activity of several GAT-1 variants [22], as well as that of other transporters, such as serotonin (SERT) [26], creatine (CRT1) [25] and glycine (GlyT2) [27]. Treatment of GAT-1WT-transfected HEK293 cells with PBA (2 mM, 24 h) produced a significant increase in the [^3^H]GABA uptake (Figure 11A).

Increased activity was also observed for most of the mutants tested, but taking into account their low baseline activity, the absolute value of GABA uptake increased only modestly. The effect of PBA was also assayed after coexpression of each mutant with the WT (in a 1:1 ratio) in order to simulate heterozygosity of patients [22]. Every combination resulted in stimulation of activity, reaching values similar to that observed for the WT (Figure 11B). Western Blot analysis of the cellular extracts indicated that the amount of protein in the mature forms of GAT-1 band increased for most of the mutants (Figure 11B,C), and PBA treatment caused an electrophoretic mobility shift of the mature band to a higher apparent Mr, while the size of the core glycosylated bands was not affected.

## 3. Discussion

In this work, we describe the pathogenic mechanism of a new variant of the *SLC6A1* gene found in two unrelated Spanish families that confers epilepsy, autism and mental retardation. The G307R variant of GAT-1 appeared de novo in both patients, leading to a nearly complete loss of [^3^H]GABA reuptake activity. Although a remarkable number of variants in this gene have been described (https://gnomad.broadinstitute.org/transcript/ENST00000287766?dataset=gnomad_r2_1) (accessed on 20 November 2022), work to determine their pathogenic mechanisms has hardly been carried out until recently, when the mechanisms of at least eight mutants were described in several separate studies [20,21,22,23,24,34]. Our work adds six novel mutants (the seventh, A288V, overlaps with the study by [22]). Knowledge of the pathogenic mechanisms of a growing number of harmful variants of *SLC6A1* may help to better classify patients and facilitate the search for individualized treatments. Interpretation of these results is now favored by the recent publication of the 3D structure of GAT-1 [10], which allows each variant to be located precisely. This breakthrough is what prompted us to carry out a bioinformatic study attempting to correlate the effects of these variants with the biophysical alterations they introduce. However, the analysis revealed great difficulty in prediction of GAT-1 trafficking to the membrane. In particular, predicted changes of folding free energy were not consistent among the different predictors, which are possibly not even qualitatively correct for membrane proteins (having been developed on proteins that exist in the aqueous solvent).

In contrast, the dynamic view of the protein native state as a statistical ensemble represented through elastic network models, though highly simplified through the harmonic approximation, allowed us to rationalize the lack of function of two mutants (G307R and A334P) that were present on the membrane, but showed reduced transport activity. These mutants are the only ones that exerted a large deformation on functional sites (in particular, the intracellular gate) and were predicted to produce a large RMSD, with low accessibility to the WT native structure.

For one mutant (G362R), the reduced transport to the membrane almost exactly explained the reduction in activity, but our analysis could not rationalize the decreased activity of the 2 (of 5) mutants that were observed on the membrane in relevant amounts. D410 is part of the intracellular gate, thanks to its ionic interaction with R44 (TM1a), and disruption of this interaction in the D410E variant abolishes transport activity, with hardly any effect on surface trafficking [22]. Therefore, the reduced activity of D410E might be explained through fine details of the atomic interactions, although our analysis predicts that this mutant is associated with a small RMSD, and that the WT structure remains accessible. Finally, the reduced activity of the mutant L460R [22] could not be explained by either the reduced traffic to the membrane, nor the predicted structural deformation.

These limitations underscore that bioinformatic methods are still far from being predictive of protein activity. Nevertheless, in combination with experiments, methods based on protein dynamics may be useful to understand the effects of mutations.

A further consideration about the GAT-1 mutant G307R is that the adjacent residues (S308, Y309 and N310) are known to be involved in intracellular gate closure. Specifically, N310 establishes interactions with W47 (TM1a) [10,35], whose geometry could be altered by the new interactions of R307, perhaps with TM2, as predicts Dynamut2 (Appendix A). Similarly to G307R, mutations of those adjacent residues S308 and Y309 are transported to the surface (although less efficiently than GAT-1 WT), but are also virtually inactive [35]. Similar results have been published for the extracellular gate, where alteration of the critical residue E451 cancels the transporter activity, but not GAT-1 trafficking to the surface [36]. Therefore, it seems that mutation of residues close to the gates has a great impact on transporter activity, but barely affects trafficking. On the contrary, positions close to the lipid bilayer, such as A128V or G550R, are misfolded and cannot escape from the ER. Indeed, similar observations have been made for the DAT transporter, where most of the residues that cause protein misfolding upon mutation (and give rise to infantile parkinsonism/dystonia) are located at the protein-lipid interface [37]. To some extent, the peripheral mutation G232V also fits this idea. Although it reaches the surface, it does so in small quantities and only in cells with low GAT-1 expression. In addition, G232 is not very close to the bilayer, being on the border between TM4 and IC2 and, thus, in a more hydrophilic environment.

Despite the efficient trafficking of GAT-1-harboring mutations close to the external and internal gates, mutation of residues placed deeper in the permeation pathway are also rejected by the quality control mechanism and remain in the ER. This is the case for mutant G297R, located in the uncoiled zone of TM6a-TM6b, and G295L, located in TM6a [22] and participating in the binding of GABA and Na^+^. Interestingly, despite this dramatic effect, G297 had one of weakest predictions of pathogenicity by the PolyPhen-2 program. A288V, which is also found in TM6a, was largely retained in the ER, although the fraction that reached the membrane was functional. Indeed, mutant A288V had a higher activity than would otherwise be expected, based on its protein concentration in the plasma membrane, which may result either from membrane saturation in the WT controls or due to a more efficient catalytic cycle of the few A288V molecules that reached the surface.

In relation to the clinical effects of the G307R variant, like many other GAT-1 variants, the patients showed MAE and intellectual disability, as well as epilepsy. Variants in *SLC6A1* were first identified in patients presenting MAE, which is characterized by a range of seizure types, including myoclonic, myoclonic-atonic, atonic and absence seizures [1,19]. Additionally, patients with MAE have variable degrees of intellectual disability and developmental delay, and some cases present with behavioural problems, namely, attention deficit, hyperactivity, aggressive behaviour, hand stereotypies and autistic features in different combinations [2,18,19,38]. Furthermore, there are descriptions of patients with autism spectrum disorder and language regression similar to our patients’ phenotype [39,40]. Finally, focusing on the treatment to improve epilepsy in our patients, administration of VPA has given a good response. These results of the treatment with VPA are similar to those obtained in patients with other *SLC6A1* variants [2,41].

Finally, regarding the effect of PBA on GAT-1 activity and expression, our results add to those obtained by Nwosu et al. [22], indicating that this is a drug that could have some utility for treatment of GAT-1 dysfunction. First, the compound enhanced the activity of GAT-1 WT, supporting that it stimulated the processes associated with trafficking, such as the folding and the glycosylation of GAT-1. While the effect on folding has been reported for many membrane proteins, its effect on glycosylation has been reported more rarely, although the creatine and serotonin transporters are more glycosylated in the presence of PBA [25,26]. There is also evidence indicating that this compound accelerates the trafficking of some secretory proteins through the Golgi in a manner dependent on the GRASP55 protein resident in this organelle [42]. This suggests a better access to Golgi-resident glycosylating enzymes of proteins in transit to the cell surface. Relative to the mutants, the absolute improvement in activity was rather small for all the assayed mutations when each one was transfected in HEK293 cells due to the marginal intrinsic activity of most of them. Yet, interestingly, PBA also increased the activity of a mutant like G307R, which, even upon reaching the membrane, is largely inactive under basal conditions, suggesting that PBA might also assist the conformational changes occurring during the alternative access transport cycle. However, it must be considered that patients are heterozygous for this locus, and a stimulatory effect of PBA was consistently seen on the WT form in “heterozygous”-transfected cells. PBA increased the transport activity independently of the mutant that was coexpressed with GAT-1 WT.

Of interest for the potential use of PBA in the treatment of some forms of epilepsy, it was approved by the FDA in 1996 for the treatment of urea cycle disorders and recently (2022) also for that of amyotrophic lateral sclerosis (https://www.fda.gov/news-events/press-announcements/fda-approves-new-treatment-option-patients-als) (accessed on 20 November 2022). Therefore, it appears that at least some of the variants that cause transporter misfolding could benefit from chemical chaperone treatments.

In summary, in this article, we provide new experimental data on the pathogenic mechanisms of seven mutations in the *SLC6A1* gene, and we try to rationalize these mechanisms with the protein structure and the bioinformatic methods currently available. In addition, we extend the previously published observations for many other GAT-1 mutants regarding the encouraging response to treatment with the chemical chaperone PBA.

## 4. Materials and Methods

### 4.1. Patients, Sample Preparation and DNA Analysis

Written informed consents for molecular analyses were obtained in public Spanish Hospitals (patient 1 in Hospital Sant Joan de Déu, Barcelona; patient 2 in Hospital Universitario Donostia, San Sebastián). Whole exome sequencing by trios was carried out, as described in Lucariello et al. 2016 [43]. We prioritized the variants using ACMG-AMP criteria, as described by Richards et al. 2015 [44]. Furthermore, we took into account the results observed in ClinVar and HGMD-Professional, which aggregate information about genomic variation and its relationship to human health, and also Varsome and Franklin, a variant knowledge community, data aggregation and variant data discovery tool [45].

The variants were validated by Sanger sequencing using the BigDye Terminator v3.1 Cycle Sequencing Kit (Life Technologies, Grand Island, NY, USA) in an Applied Biosystems 3730/DNA Analyzer (Applied Biosystems—Life Technologies—Grand Island, NY, USA). The subsets of primers used were SLC6A1_919F (catggaaaagttcctggatagc) and SLC6A1_919R (cctaacacatgagtgtgggatg).

### 4.2. Site-Directed Mutagenesis

The rat GAT-1 cDNA was previously cloned in the *Bam*H1/*Eco*RI sites of pCDNA3 containing the HA tag. All GAT-1 mutations of interest were generated using Quickchange Site-directed Mutagenesis Kit Lightning (Agilent), using the following oligonucleotides: A128V-Fw gtgtgggcctcgtggcagctgtgct; A128V-Rv agcacagctgccacgaggcccacac; G232V-Fw tttctgcatctggaaggttgttggatggactggaa; G232V-Rv ttccagtccatccaacaaccttccagatgcagaaa; A288V-Fw gtgtggcttgacgtcgccacccagatc; A288V-Rv gatctgggtggcgacgtcaagccacac; G297R-Fw cttcttctcctacaggctgggcctggg; G297R-Rv cccaggcccagcctgtaggagaagaag; G307R-Fw ggtccctgattgctctgagaagctacaactctttc; G307R-Rv gaaagagttgtagcttctcagagcaatcagggacc; A334P-Fw ctcctgcaccagcatgtttcccggattcgt; A334P-Rv acgaatccgggaaacatgctggtgcaggag; G550R-Fw catggtgctcatccccaggtacatggcttacat; G550R-Rv atgtaagccatgtacctggggatgagcaccatg. The products were confirmed by DNA sequencing.

### 4.3. Primary Cultures

Brain cortex from 18-day-old rat fetuses was cultured as described elsewhere [46]. For transfection and immunofluorescence assays, the cells were grown in 24-well plates coated in poly-L-lysine (13 μg/mL) at a density of 30,000 cells/well.

### 4.4. Transfection

HEK293 cells were transfected in p24 or p6 multiwell plates with Turbofect, following the manufacturer’s instructions. Primary neurons were transfected using calcium phosphate, following the manufacturer’s instructions (ProFection^®^ Mammalian Transfection System, Promega, Madison, WI, USA).

### 4.5. Immunofluorescence

Primary cultures of transfected neurons or transfected COS7 cells were fixed in 4% paraformaldehyde/PBS, blocked, permeabilized and incubated with primary and secondary antibodies, as described previously [46]. The coverslips were mounted in fluoromount-G for analysis. Images were collected on an LSM510 confocal microscope coupled to an Axiovert200 M inverted microscope (Zeiss, Jena, Germany).

### 4.6. Radioactive GABA Transport

The radioactive [^3^H]GABA uptake assay was performed in transfected HEK293 cells grown in 24-well plates. Cells were first incubated in PBS containing 10 mM glucose and 0.1 µCi [^3^H]GABA for 10 min. Then, cells were washed with PBS and lysed with 250 µL of 0.2 N NaOH for 1 h. Next, 150 µL of cell lysate was added to 1.5 mL of scintillation liquid, and radioactivity was determined using a Packard β-scintillation QuantaSmart counter. After, 5–10 µL of the lysate was used for protein determination by the Bio-Rad Protein Determination kit, with bovine serum albumin (BSA) as the standard. The untransfected condition was taken as baseline and subtracted from both the WT and mutant conditions. For PBA treatments, the compound was added 6 h after transfection and maintained in the culture for 24 h until the transport assay was carried out.

As described previously [47], cell surface proteins of transfected HEK293 cells were labelled for 20 min at 4 °C by incubating them in a solution containing the non-permeable sulfo-NHS-SS-biotin reagent (1 mg/mL in PBS). After exhaustive washing, cells were lysed for 30 min in 1 mL of lysis buffer (150 mM NaCl, 5 mM EDTA, 50 mM HEPES-Tris, 0.25% sodium deoxycholate, 1% Triton X-100, 0.1% SDS [pH 7.4]), and the lysate was cleared by centrifugation at 14,000× *g* for 10 min. Biotinylated proteins were finally recovered by incubating the cleared lysate for 2 h at room temperature with streptavidin-agarose beads. After washing, protein bound to the beads was eluted in 2 × Laemmli sample buffer, separated by SDS-PAGE and analyzed by western blot. Biotinylated proteins were visualized with the corresponding antibodies.

### 4.7. Electrophoresis and Immunoblotting

SDS-PAGE was performed on 7.5% polyacrylamide gels in the presence of 2-mercaptoethanol. After electrophoresis, protein samples were transferred to nitrocellulose membranes, as described previously [46]. Membranes were probed with anti-HA or anti-mCherry and Calnexin, visualized by ECL and quantified by densitometry (Image Lab 6.0).

### 4.8. Prediction of the Effect of Mutations on Stability and Structure

We used a number of bioinformatic tools in order to predict the effect of mutations on protein stability: SDM [48], which predicts the native free energy based on a statistical energy function; mCSM [49], which uses graph-based signatures to represent the wild-type structure and machine learning to predict the effect of mutations on protein stability; DUET [50], which obtains consensus predictions based on two previous predictors; EnCOM [51], which predicts vibrational entropy changes based on the elastic network model; Dynamut [52], which integrates DUET and EnCOM; Dynamut 2 [53], which is a further development of Dynamut; MAESTROweb [54], which implements a multi-agent machine learning system for predicting folding free energy changes upon mutation; INPS-3D [55], which estimates the free energy of the native state based on a contact potential; POPMusic [56], which predict the changes in thermodynamic and thermal stability by using a linear combination of statistical potentials, whose coefficients depend on the solvent accessibility of the mutated residue; and DeltaGREM [57], which adopts a contact energy function in order to predict the difference of free energy between the native state represented in the PDB and the most stable state between the unfolded state and the ensemble of incorrectly folded compact conformations [58]. The precise localization of each variant was visualized using Chimera 1.16 software (Resource for Biocomputing, Visualization and Informatics, University of California, San Francisco, CA, USA) [59].

### 4.9. Analysis of Protein Dynamics with the Torsional Network Model (TNM)

We applied an elastic network model [60] in torsion angle space (torsional network model; TNM [32] in order to predict the dynamics of GAT-1 in the native state represented by the experimental structure in the PDB 7sk2. The TNM is a structure-based model that uses all protein heavy atoms to compute the kinetic and potential energy, approximating the latter as a quadratic function (harmonic approximation), and freezes all degrees of freedom besides the phi and psi backbone torsion angles, which allows fast analytic computations of all Boltzmann averages of interest as Gaussian averages over the independent normal modes.

By applying the TNM, we predicted the structural change caused by GAT-1 mutations as the linear response of the elastic network to the perturbation induced by the mutation [31]. We also predicted through the TNM the dynamical couplings between the mutated sites and the functional sites of the protein [33].

### 4.10. Statistical Analysis

Quantitative data were analyzed using GraphPad Prism 7.0. Two-tailed paired or unpaired Student’s *t* test was used to calculate the *p*-values, with a cutoff for significance set at 0.05. The types of the statistical tests, the sample size, exact *p*-values and statistical significance are reported in the figures and corresponding figure legends. Data shown in graphs represent the mean ± standard error of the mean (SEM).

## Figures and Tables

**Figure 1 ijms-24-00955-f001:**
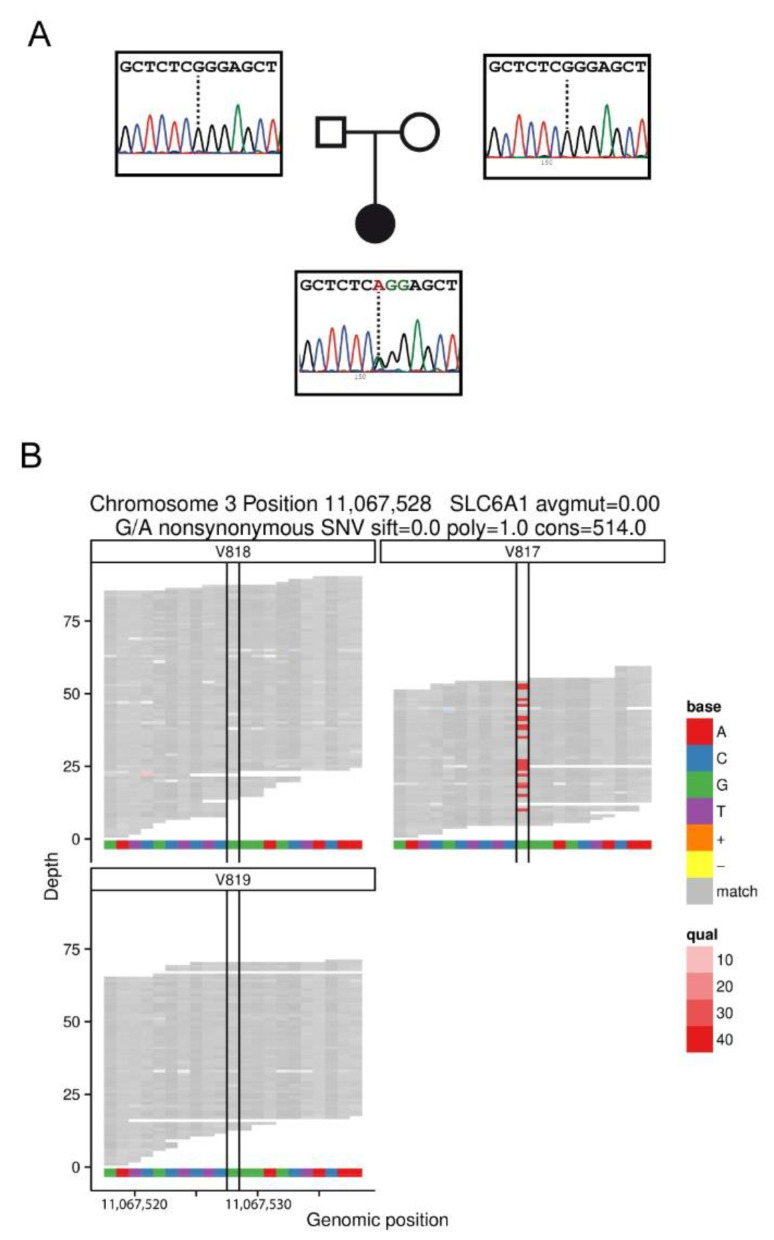
(**A**) Pedigree and Sanger sequencing of the family 1 trio. (**B**). Integrated genomics viewer (IGV) screenshot from WES coverage data of point mutation c.919G>A (p.Gly307Arg) in trio family from patient 1 (genomic location chr3:11067528; V817 index case; V818 father; V819 mother).

**Figure 2 ijms-24-00955-f002:**
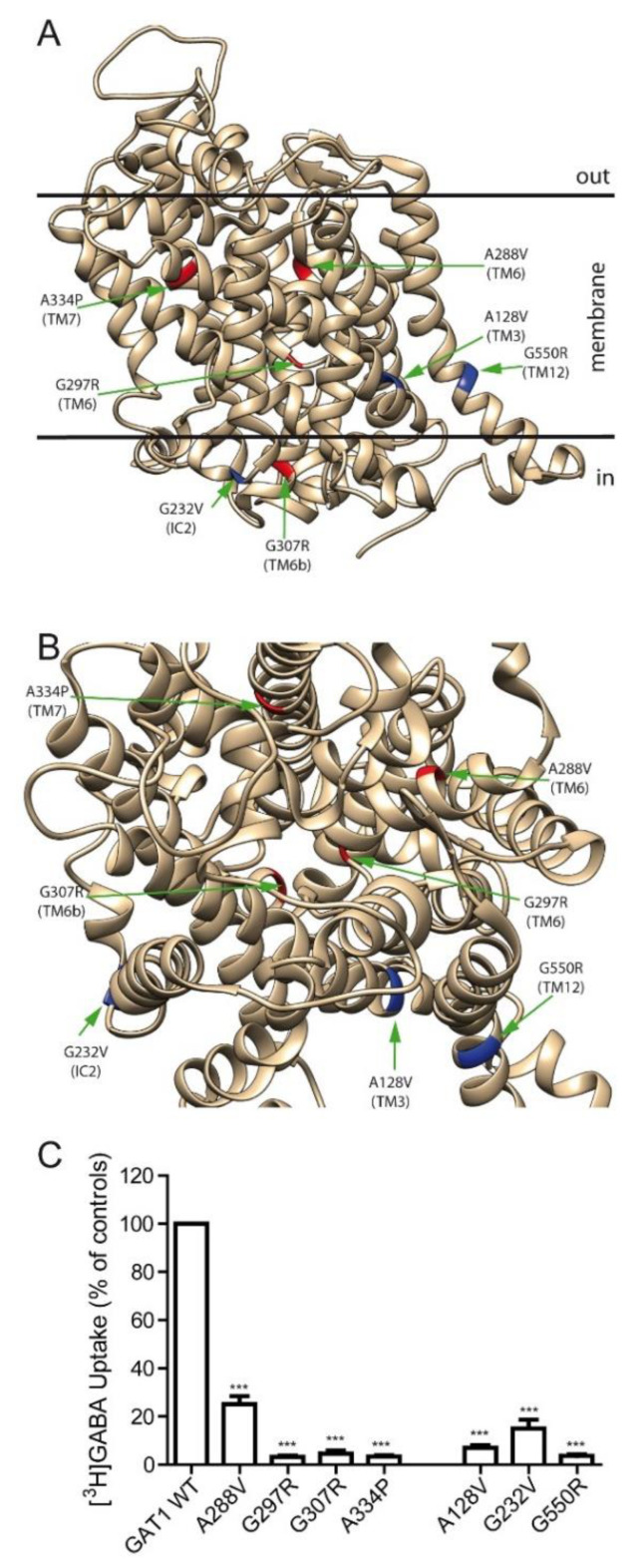
Effect of GAT-1 mutations on its activity. (**A**,**B**) Lateral (**A**) and top (**B**) view of the ribbon diagram of GAT-1 (PDB: 7sk2) indicating the location of the variants studied in this article. Peripheral mutations are indicated in blue, and those located within the permeation pathway in red. The position of the membrane is indicated in the lateral view by two black lines. (**C**) GABA uptake activity measured in transfected HEK293 cells. Cells were transfected and 48 h later were incubated for 10 min in the presence of 0.1 µCi of [^3^H]GABA, and the incorporated radioactivity was determined in a scintillation counter. Values represent the percentage of the activity observed in the wild-type control (GAT-1 WT) after correction for the background observed in mock-transfected cells. Values are the means ± SEM of three experiments. Average CPM values per well were 5387 for the controls and 254 for the mock-transfected cells. Values are the mean ± SEM of four experiments. ***: *p* < 0.001, paired *t*-test.

**Figure 3 ijms-24-00955-f003:**
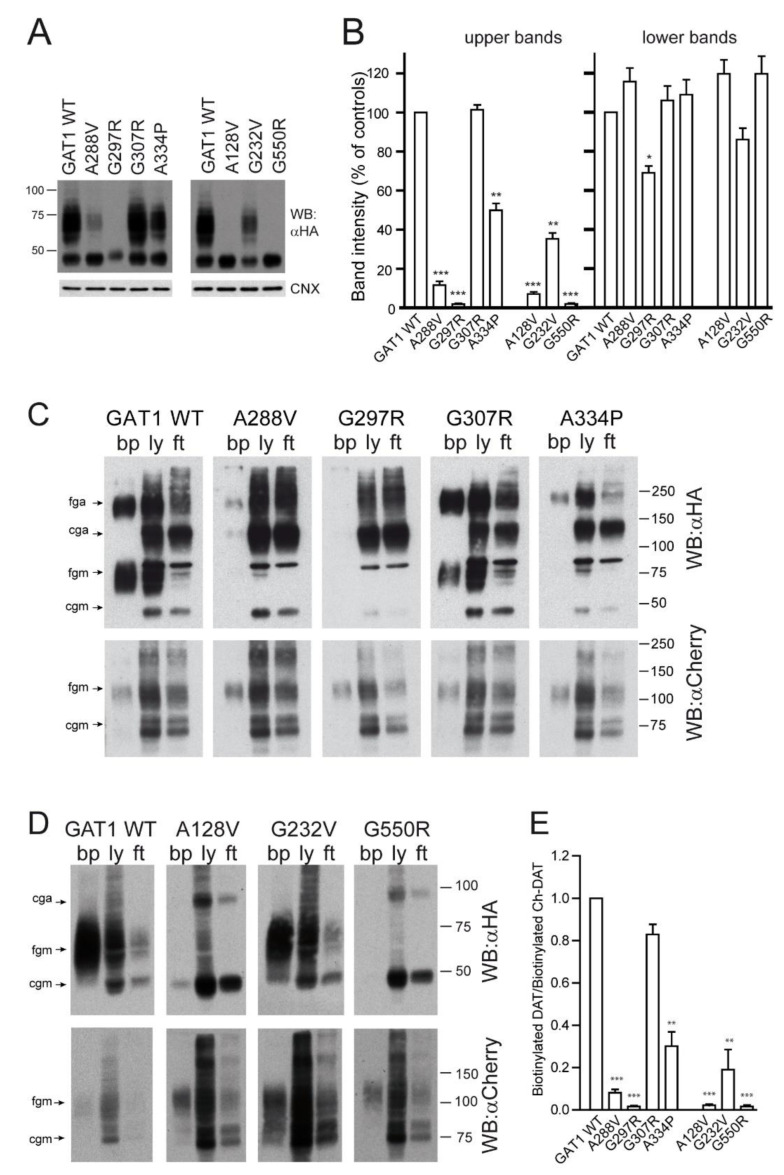
Expression of GAT-1 variants in transfected HEK293 cells. (**A**). HEK293 cells were transfected with the indicated variants of GAT-1 as indicated in Figure 2. Cell were lysated and samples were analyzed by western blot using anti-HA antibody and anti-calnexin as a loading control. (**B**). Densitometric analysis of the fully (upper) and the core glycosylated (lower) bands expressed as percentage of the GAT-1 WT variant. Values are the mean ± SEM of four experiments. *: *p* < 0.01; **: *p* < 0.005; ***: *p* < 0.001, paired *t*-test. (**C**,**D**). Biotinylation assay. HEK293 cells were co-transfected with mCherry-DAT plus either GAT-1 WT (control) or the indicated variants. Two days later, cell surface proteins were labeled with sulfo-NHS-biotin. Biotinylated proteins were separated from non-biotinylated ones with streptavidin-agarose beads and eluted from the beads. Samples of biotinylated protein (bp), lysate (ly) and flow through (ft) were resolved by SDS-PAGE and analyzed by immunoblotting with anti-HA antibodies, and reprobed with anti-mCherry antibodies. The different bands correspond to core glycosylated protein, either monomeric or aggregated (cgm and cga), and fully glycosylated protein (fgm, fga). (**E**) Histograms corresponding to densitometric analysis of biotinylated proteins. Values correspond to the ratio between the densitometric adjusted volumes for the biotinylated bands of HA-GAT-1 (monomers plus aggregated proteins) divided by the biotinylated band for mCherry-DAT (monomers). Values represent the mean ± SEM of four experiments after normalization to the ratio obtained for the controls (GAT-1 WT). **: *p* < 0.005; ***: *p* < 0.001, paired *t*-test.

**Figure 4 ijms-24-00955-f004:**
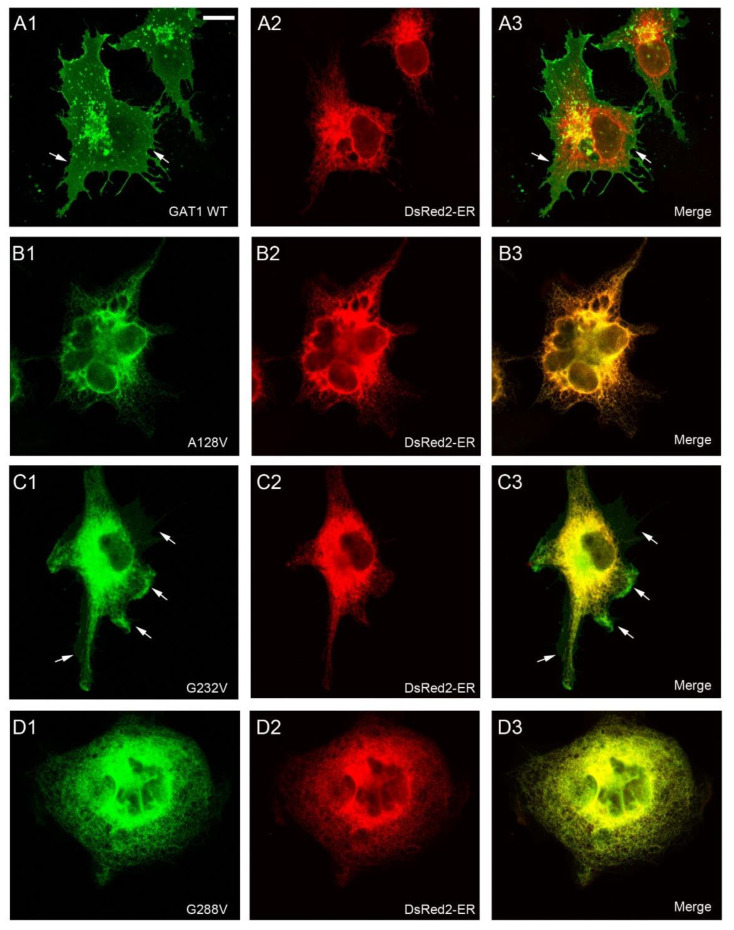
Subcellular distribution of GAT-1 variants in transfected cells. COS7 cells were transfected with expression vectors for HA-GAT-1 WT (**A1**–**A3**) or the indicated HA-tagged mutants: A128V (**B1**–**B3**); G232V (**C1**–**C3**); G288V (**D1**–**D3**). Transfection mixture also contained the expression vector for the fluorescent protein DsRed2-ER, which localized to the ER. Two days later, cells were fixed with 4% paraformaldehyde, immunostained with anti-HA plus anti-mCherry antibodies, and analyzed by confocal microscopy. Note the presence of membrane sheets in some cells (arrows), although in some cases (G232V) they are very faint. Scale bar: 15 µm.

**Figure 5 ijms-24-00955-f005:**
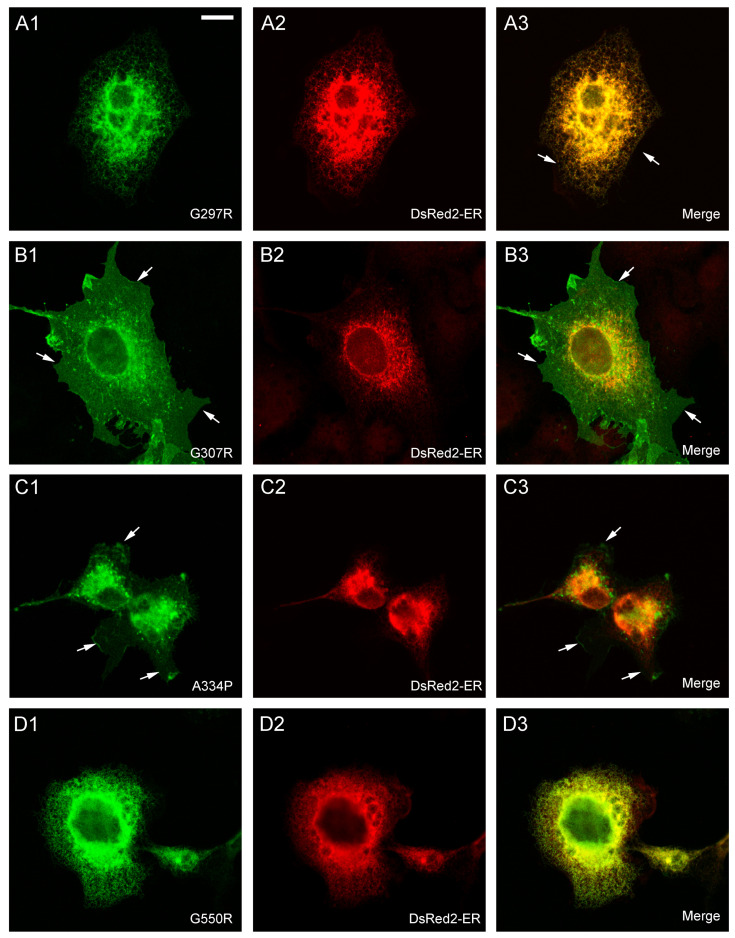
Subcellular distribution of GAT-1 variants in transfected cells. COS7 cells were transfected with expression vectors for the indicated HA-tagged mutants: G297R (**A1**–**A3**); G307R (**B1**–**B3**); A334P (**C1**–**C3**); G550R (**D1**–**D3**). Transfection mixture also contained the expression vector for the fluorescent protein DsRed2-ER, which localized to the ER. Two days later, cells were fixed with 4% paraformaldehyde, immunostained with anti-HA plus anti-mCherry antibodies and analyzed by confocal microscopy. Note the presence of membrane sheets in some cells (arrows), although in some cases (A334P) they are very faint. Scale bar: 15 µm.

**Figure 6 ijms-24-00955-f006:**
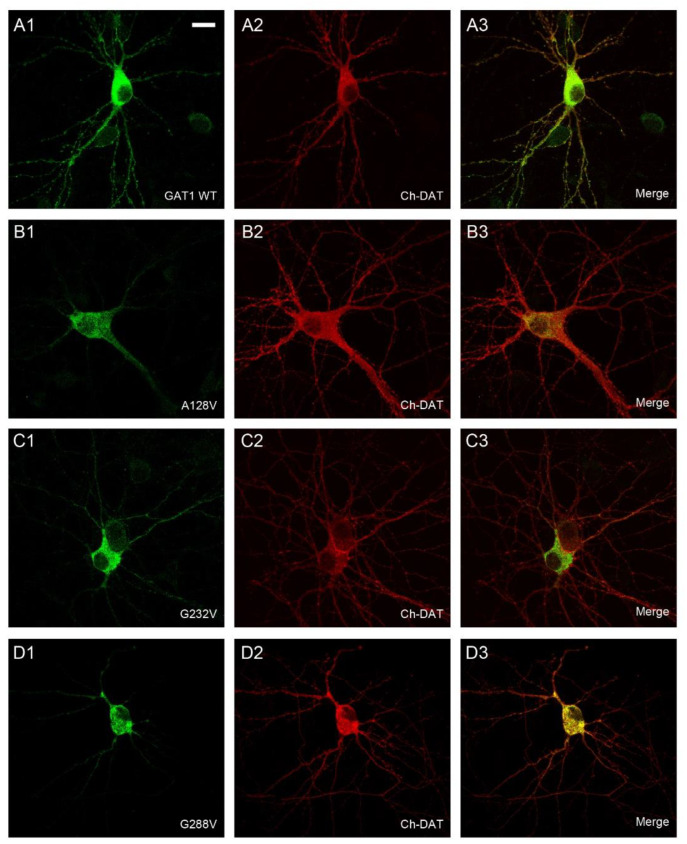
Subcellular distribution of GAT-1 variants in transfected neurons. Primary cultures of rat cortical neurons were co-transfected at 13 DIV with mCherry-DAT plus the indicated variants of HA-GAT-1: HA-GAT1 WT (**A1**–**A3**); A128V (**B1**–**B3**); G232V (**C1**–**C3**); G288V (**D1**–**D3**). 48 h later, cells were fixed and incubated with anti-HA and anti-mCherry primary antibodies followed by Alexa Fluor-labeled secondary antibodies. Images collected by confocal microscopy correspond to the distribution of HA-GAT-1 variants (green channel; **A1**,**B1**,**C1**,**D1**), mCherry-DAT (red channel; **A2**,**B2**,**C2**,**D2**) or the merge (**A3**,**B3**,**C3**,**D3**). Scale bar: 15 µm.

**Figure 7 ijms-24-00955-f007:**
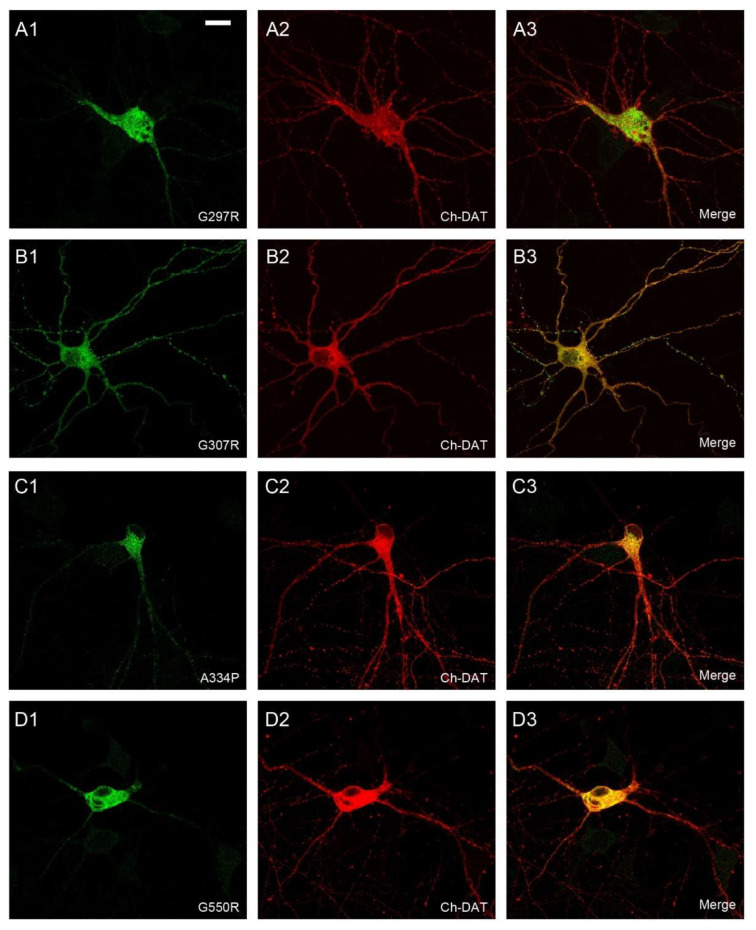
Subcellular distribution of GAT-1 variants in transfected neurons. Primary cultures of rat cortical neurons were co-transfected at 13 DIV with mCherry-DAT plus the indicated variants of HA-GAT-1: G297R (**A1**–**A3**); G307R (**B1**–**B3**); A334P (**C1**–**C3**); G550R (**D1**–**D3**). 48 h later, cells were fixed and incubated with anti-HA and anti-mCherry primary antibodies followed by Alexa-labeled secondary antibodies. Images collected by confocal microscopy correspond to distribution of HA-GAT-1 variants (green channel; **A1**,**B1**,**C1**,**D1**), mCherry-DAT (red channel; **A2**,**B2**,**C2**,**D2**) or the merge (**A3**,**B3**,**C3**,**D3**). Scale bar: 15 µm.

**Figure 8 ijms-24-00955-f008:**
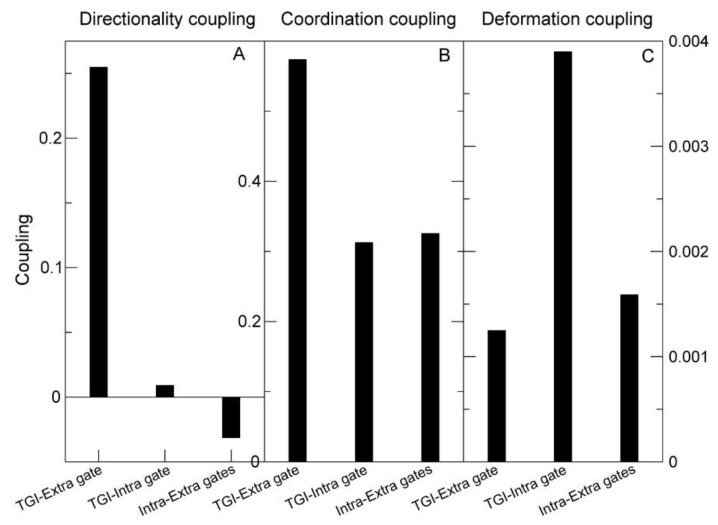
Directionality coupling (**A**), coordination coupling (**B**) and deformation coupling (**C**) between three pairs of functional sites (horizontal axis): extracellular gate with tiagabine binding site (TGI) (first bar), intracellular gate with TGI (second bar) and intracellular gate with extracellular gate (third bar). The couplings are calculated with PDB:7sk2, which is an inward-open conformation in complex with the inhibitor tiagabine. Note that the TGI, which is thought to overlap the GABA binding site, is strongly co-directional and coordinated with the extracellular gate, but has low co-directionality and coordination with the intracellular gate. On the contrary, the TGI and the intracellular gate can strongly perturb each other (**C**).

**Figure 9 ijms-24-00955-f009:**
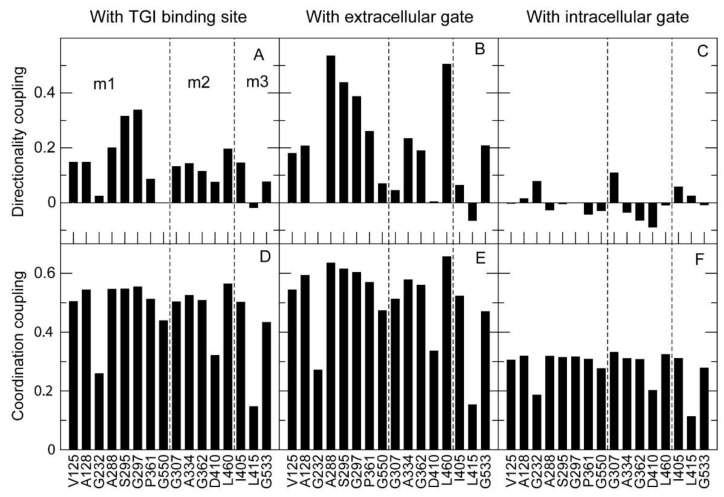
Directionality coupling (**A**–**C**) and coordination coupling (**D**–**F**) between the 16 mutated sites (horizontal axis) and three functional sites of GAT-1: the tiagabine binding site (TGI) (first column, **A**,**D**), the extracellular gate (second column, **B**,**E**) and the intracellular gate (third column, **C**,**F**). Mutants are shown in three groups separated by dashed lines: pathogenic mutants that prejudice transport to the membrane (m1); pathogenic mutants that do not prejudice transport to the membrane (m2); and non-pathogenic mutants (m3). Note that most sites are positively co-directional and strongly coordinated with the tiagabine binding site, which is thought to overlap the GABA binding site.

**Figure 10 ijms-24-00955-f010:**
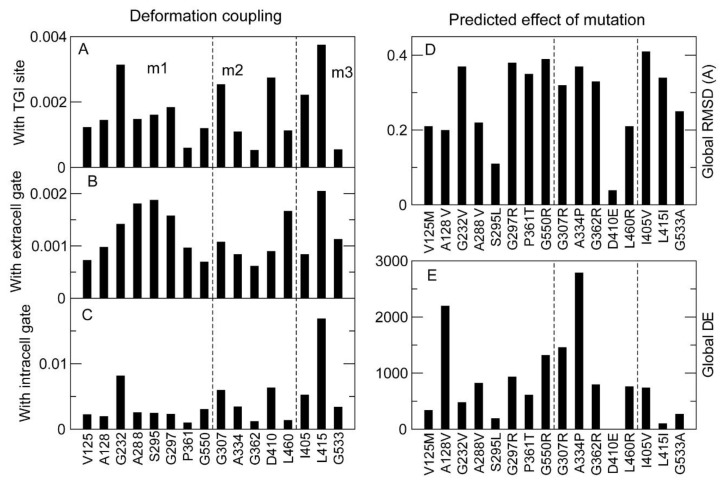
Deformation coupling between the 16 mutated sites (horizontal axis) and three functional sites: the tiagabine binding site (TGI) (**A**), the extracellular gate (**B**) and the intracellular gate (**C**). (**D**,**E**) Predicted structural effect of specified mutations on the global structure of the protein, assessed in terms of predicted RMSD between the native structure of the wild-type and the mutant (**D**), and predicted harmonic energy barrier ΔE between the two structures (**E**), which informs of the accessibility of the native structure of the WT in the native protein. The mutants are shown in three groups separated by dashed lines: pathogenic mutants that prejudice transport to the membrane (m1); pathogenic mutants that do not prejudice transport to the membrane (m2); and non-pathogenic mutants (m3).

**Figure 11 ijms-24-00955-f011:**
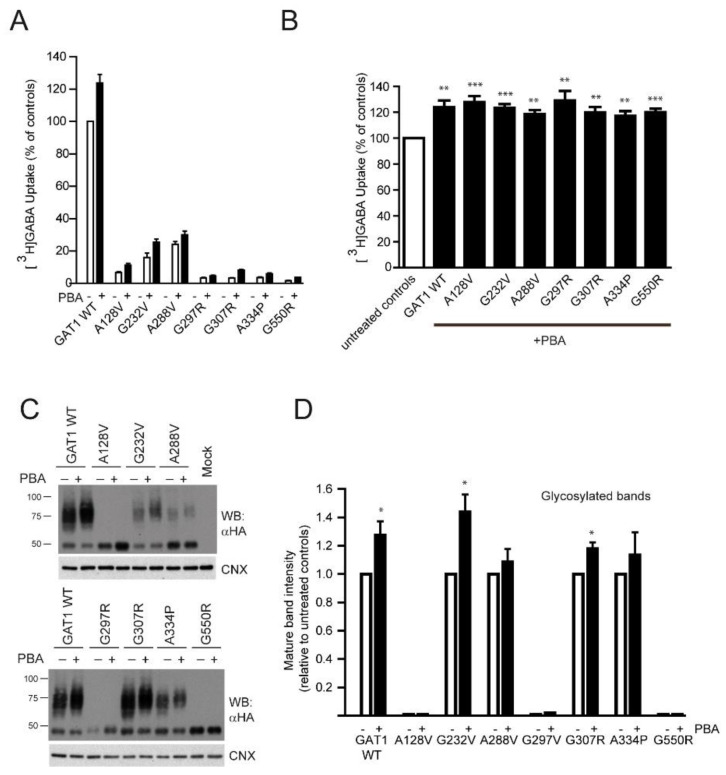
Effect of 4-phenylbutyrate on the activity and expression levels of GAT-1. (**A**). HEK293 cells were transfected with 0.5 µg of the expression vectors for the indicated variants of GAT-1, and 24 h later were treated with 2 mM PBA for 24 additional hours. Then, cells were incubated for 10 min in the presence of 0.1 µCi of [^3^H]GABA and the incorporated radioactivity was determined in a scintillation counter. Values represent the percentage of the activity observed in the controls (GAT-1 WT) after correction for the background observed in mock-transfected cells and represent the mean ± SEM of three triplicate experiments. **: *p* < 0.005; ***: *p* < 0.001, paired *t*-test. (**B**). HEK293 cells were transfected with 0.25 µg of the expression vectors for the indicated variants of HA-GAT-1 plus 0.25 µg of HA-GAT-1 WT. Uptake assays were performed and processed as indicated in (**A**). (**C**). Samples of transfected cells were analyzed by western blot using anti-HA antibody and reprobed with anti-calnexin. (**D**). The intensity of the mature glycosylated band was measured by densitometry. Values represent the mean ± SEM of four experiments after normalization to the ratio obtained for the untreated controls. *: *p* < 0.01, paired *t*-test.

**Table 1 ijms-24-00955-t001:** Clinical characteristicsof patients included in this study.

Patient no.	Gender/ Age at Inclusion	Family History	Cognition before Epilepsy Onset	Age at Epilepsy Onset	Seizure Type	EEG	Cognition after Seizure Onset	Behavioral Problems	Neurological Findings	Effective AED	Mutation
**1**	F/47 y	None	Moderate ID	5 y	Absences atonic, eyelid myoclonia	Generalized spike and wave discharges at 3 Hz	Moderate ID	Autism Spectrum Disorders	Hypotonia	Sz free on VPA + CLZ	NM_003042.3: c.919G>A (NP_003033.3: p.Gly307Arg) de novo
**2**	M/2.5 y	None	Moderate ID	15 m	Absences, eyelid myoclonia	Slow background rhythm, occipital spike-wave activity.	Moderate ID	Autism Spectrum Disorders	Hypotonia, Midline stereotypies	Sz free on VPA + CLZ	NM_003042.3: c.919G>A (NP_003033.3: p.Gly307Arg) de novo

**Table 2 ijms-24-00955-t002:** Fraction of GAT-1 protein reaching the membrane in experimentally verified variants of the SLC6A1 gene, and ΔΔG calculated by different bioinformatic methods. Negative values indicate destabilization of the protein. Correlation coefficients with membrane fraction are also calculated. Accessibility column corresponds to solvent accessibility calculated by Pop_Music. PrDa, probably damaging; Ps-Da, possibly damaging; Beni, benign.

Mutation	Membr Fraction (%)	Dynamut2	Dynamut	ENCoM	mCSM	SDM	DUET	Maestro Web	Pop_ Music	INPS _3D	Delta GREM	-TNM _RMSD	-TNM_DE	Accessi-bility	Hum_Div	Hum_Var
V125M	25	−0.39	0.68	−0.166	−0.292	−0.63	−0.2	−0.318	−1.19	−1.164	−0.905	−0.21	−339	0.5	Pr-Da	Pr-Da
A128V	2	−0.25	−0.298	−0.386	−0.469	−1.03	−0.33	−0.463	−0.73	−0.509	−0.914	−0.2	−2200	0	Pr-Da	Pr-Da
G232V	20	−0.58	−0.67	0.071	−0.057	0.09	0.311	0.154	−0.76	−1.391	−0.121	−0.37	−480	40.23	Pr-Da	Pr-Da
A288V	10	−0.51	−0.092	−0.581	−0.169	−1.03	−0.007	−0.515	−0.59	−1.424	−0.381	−0.22	−823	0	Pr-Da	Pr-Da
S295L	2	−0.65	−0.023	0.073	−0.286	2.07	0.542	−0.55	−0.67	−0.703	0.04	−0.11	−195	15.76	Pr-Da	Pr-Da
G297R	0	−0.19	0.069	−0.744	−0.545	−3.1	−0.865	0.0178	−2.23	−0.622	−0.85	−0.38	−935	7.06	Pr-Da	Pr-Da
G307R	80	−0.37	−0.763	−0.8	−0.763	−1.89	−0.72	−0.395	−0.16	−0.778	−0.518	−0.32	−1460	1.91	Ps-Da	Beni
A334P	35	0.59	0.383	−0.188	0.035	−4.45	−0.643	−0.0535	−1.87	−1.971	−2.444	−0.37	−2790	2.42	Pr-Da	Pr-Da
P361T	5	−1.17	−0.116	−0.209	−1.284	0.37	−0.815	−0.483	−0.63	−0.9492	0.5486	−0.35	−613	0.86	Pr-Da	Pr-Da
G362R	50	−0.86	−0.548	−0.641	−0.938	−1.188	−0.862	0.096	−1.32	−0.636	0.491	−0.33	−797	3.5	Pr-Da	Ps-Da
D410E	75	−0.53	0.076	−0.459	−0.594	0.03	−0.351	−0.5	−0.11	−0.279	−1.245	−0.039	−5.29	65.67	Pr-Da	Ps-Da
L460R	75	−1.38	−1.097	0.109	−1.427	−3.02	−1.524	0.278	−1.53	−1.769	0	−0.21	−763	0.58	Pr-Da	Pr-Da
G550R	2	−0.86	0.738	−0.067	−0.897	0.86	−0.444	−0.366	−0.25	−0.313	0.551	−0.39	−1320	41.07	Ps-Da	Beni
Correlation with membrane fraction	1	−0.119	−0.494	−0.19	−0.315	−0.349	−0.466	0.279	0.116	−0.145	−0.173	0.26	0.082	0.134	n.a.	n.a.

## Data Availability

The data presented in this study are available on request from the corresponding author.

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
