# Peer review of "Experimental and Bioinformatic Insights into the Effects of Epileptogenic Variants on the Function and Trafficking of the GABA Transporter GAT-1"

_ijms, 2023, doi:10.3390/ijms24020955_

Round 1
Reviewer 1 Report
The work entitled “The effect of epileptogenic mutations on the function and trafficking of the GABA transporter GAT-1” is a study of the effect of mutations in GABA transporter in activity and in intracellular trafficking-i.e. the ability of the mutants to reach the plasma membrane.
The main conclusions of the authors are
1) The mutants that are close to the gates of the transporter (G307R and A334P) can be glycosylated with variable efficiency and reach the membrane, albeit inactive. Mutants located in the center of the permeation pathway (G297R) or close to the lipid bilayer (A128V, G550R) are retained in the endoplasmic reticulum.
2) The addition of a chemical chaperone (4-phenyl- butyric acid, PBA) that is considered to improve the protein folding, increased the activity of GAT-1WT, as well as most of the assayed variants,
Major considerations
The article intends to reveal the relationship between mutations encountered in patients and the lack of transporter activity either due to ER retention, or to a misfolded protein reaching the plasma membrane. This difference is based in western quantification of bands in western blot that are not normalized to the actual expression levels of the mutant variants in each case, which may distort the interpretation of the results. The article is written in a way that sometime is cumbersome read, as the order in which results are presented lead to wrong conclusions. For instance, in figure 2 only high molecular weight isoform is considered for the quantification of the protein, but no rational for such choice at this time is provided. Why do you assume at this point that only fully glycosylated form is the mature protein? There are many examples of mature proteins (even secreted ones) with heterogeneity of glycosylation isoforms. Possibly inverting (or merging) figures 2 and 3 would clarify the criteria of quantification, otherwise one may think that the quantification of upper smeared band only may distort the interpretation of the results.
1) Please, consider rewriting the article in different order or merging figures
2) Indicate the regions in the structure in Figure 1 to see where mutations are located. Also, indicate the position of the plasma membrane to understand which mutations are located in each side of it.
3) Please, explain how did you normalize the quantification of active proteins for variations in the expression levels of the transporters
4) Indicate the molecular weight of the non-glycosylated form to interpret the size increase in both glycosylated isoforms in the 3 sequons.
5) Why protein size is different in figures 2 and 3?
6) Figure 4: size bar? Pictures have the same magnification? Why the expression is distorting the shape of the cells? The membrane vs ER localization in this figure is not convincing. Please show a co-localization with ER and membrane markers here, as the proper localization and quantification at this point is key for the interpretation of the results
7) Figure 10. Why do you think there is an increase in size of the glycosylated isoform upon treating cells with pba?
8) Materials and methods line 587: please, show the data. The transport reactions usually have high background levels and a lot of noise. The readers will appreciate to know the actual radioactivity measured in cpm, as well as that of control.
Minor considerations
· At line 95 did you meant G307R?
· Figures quality in the pdf are so poor that is hard to even follow the mutant labelling in the figures
· Line 197 explain which resin have you used
· Line 223 there is no fga in the figure
· Figure 6, no scale bars are seen in the figure, please add
Materials and methods line 611: Indicate antibody titer
Author Response
We acknowledge the constructive comments of reviewer, and will address them here:
Please, consider rewriting the article in different order or merging figures.
We agree with this suggestion, and have merged the previous Figures 2C and 3 into the new Figure 3. We have also included quantification of the immature protein bands and rewritten the text to discuss transporter activity, since we show the upper band to be the one that is located on the surface and, therefore, in contact with the substrate.
Indicate the regions in the structure in Figure 1 to see where mutations are located. Also, indicate the position of the plasma membrane to understand which mutations are located in each side of it.
We imagine that the reviewer refers to Figure 2. We have changed the display of the figure, showing a side view of the transporter and another from the top, so that it is better observed that the mutations in the permeation zone (in red now) are towards the interior of the structure, while peripheral residues (in blue) are facing outwards. In the lateral view the position of the membrane has been traced. The Figure 2 legend has been modified accordingly.
Please, explain how did you normalize the quantification of active proteins for variations in the expression levels of the transporters.
This is a difficult point. In parallel to the uptake assay we always performed a western blot to be sure that the protein was expressed and the lack of activity was not due to inadequate transfection. However, we do not have any other marker that allows for quantitative normalization. It would not be correct to normalize by the quantitative data from the western, since the amount of protein depends directly on the protein stability, which is lower in the mutants that are retained in the ER. On the other hand, none of the similar reports, either for the GABA transporter or for those of glycine, serotonin, or dopamine that we have read use this type of normalization.
Indicate the molecular weight of the non-glycosylated form to interpret the size increase in both glycosylated isoforms in the 3 sequons.
The use of the three sequons of GAT-1 was shown by Cai et al. 2005, implying a decrease in the core glycosylated form of ~7 Kda. Therefore, the apparent mobility of the peptide moiety is about 40 kDa (although this varies among laboratories due to the characteristics of gels, markers, or handling of samples). We have clarified this in the description of figure 3 (l. 240-242). Glycans are added in the Golgi on the core-glycosylated forms by a set of competing glycosyl transferases. For this reason, the population of glycans occurring at a given site is often not homogeneous; a particular site of N-glycosylation may be occupied by a number of structurally distinct glycans - a condition referred to as microheterogeneity- and is responsible of the smeared bands observed for glycoproteins.
Why protein size is different in figures 2 and 3?
The differences are due to the fact that in biotinylation experiments, for unknown reasons, there is a greater tendency for proteins to aggregate. Aggregated forms are less frequently seen in westerns. However, the size of the monomeric forms is the same in both figures.
Figure 4: size bar? Pictures have the same magnification? Why the expression is distorting the shape of the cells? The membrane vs ER localization in this figure is not convincing. Please show a co-localization with ER and membrane markers here, as the proper localization and quantification at this point is key for the interpretation of the results
Figure 4 has a scale bar, and the magnification is the same for all of them. COS cells are very heterogeneous in shape and size and that is what produces the impression that the magnification is different. These differences are not due to transfection itself but to this heterogeneity of cell sizes. In any case, figure 4 has been replaced by figures 4 and 5 in the new version, showing colocalization with an ER marker. Colocalization with a membrane-localized protein was included in Supplementary Figure 1. We estimate that the Cherry-DAT distribution gives a clearer membrane pattern than any other marker, such as the Na/K ATPase, which has been used on occasions to label the membrane. Regarding the quantitative value of the microscopy data, we think that it is much lower than that obtained biochemically by biotinylation (Figure 3) and for this reason they are not included.
Figure 10. Why do you think there is an increase in size of the glycosylated isoform upon treating cells with pba?
We have added a sentence about that in the Discussion (lines 691-696). Both serotonin and creatine transporters increase their glycosylation in the presence of PBA. Moreover, this compound increases the rate of secretion of several proteins, suggesting that the Golgi performs its job more efficiently in the presence of PBA, perhaps because proteins reach this organelle “better folded” and therefore more accessible to the competing glycosylating enzymes. However, this is only a hypothesis, and we cannot discard other possibilities.
Materials and methods line 587: please, show the data. The transport reactions usually have high background levels and a lot of noise. The readers will appreciate to know the actual radioactivity measured in cpm, as well as that of control.
These data have been included in the legend to figure 1. In GABA uptake assays, the background is quite low since HEK293 cells don’t have endogenous transporters for GABA. Background is an issue when testing amino acid transporters that have multiple uptake systems, but not for GABA in HEK293 cells.
Minor considerations
At line 95 did you meant G307R? Yes, this has been amended (now line 97).
Figures quality in the pdf are so poor that is hard to even follow the mutant labelling in the figures Sorry for that. We submitted high quality figures, and we are not sure why they have not been made available to the reviewers.
Line 197 explain which resin have you used The sentence (now line 263) has been amended. The use of streptavidin-agarose beads is also indicated in Methods (line 778)
Line 223 there is no fga in the figure. The figure has been amended
Figure 6, no scale bars are seen in the figure, please add . Our final version had the bar. In any case we have made them thicker throughout the paper.
Reviewer 2 Report
Firstly, I would like to congratulate the authors on their interesting work.
However, I would like to make some suggestions regarding the article:
1) It would be interesting if the authors could include the type of research design on the title.
2) The abstract should mention the research goals.
3) L37 : I would advise removing “reviewed in” and only citing the reference.
4) L82-93: The authors should clearly state in this paragraph what is the research question? What was the main goal of the research? What were the secondary goals? And why is this research relevant? Also, it seems that along these lines the authors are describing the findings of the research. If this is the case, it is better than the authors rewrite this paragraph and add the findings of the research to the results/discussion section. However, if those are findings from other studies, appropriate references should be added.
5) L130 : Do the authors mean the absence of Landau reflex?
6) L546: In the materials and methods section, the authors should explain and further describe:
a) Study design
b) The authors should describe in detail which statistical analyzes were performed. They should also report confidence intervals and p-values, as well as which parameters were analyzed and how. They should also describe the computer program used to carry out the analyzes.
c) Timeframe of the study
d) Setting of the study. Was it conducted in a public hospital? Private hospital? Clinic?
e) A table with baseline characteristics of the two cases should be provided.
f) The authors should provide further clinical and epidemiological information about the patients.
g) Which type of patient population is this? Which is the age of epilepsy diagnosis? Which drugs are they on? Were other AED tried? What is the frequency of the seizures per day? Do they have more than one type of seizure? How many drugs were tried? Did they have status epilepticus? How many years of following-up do these patients have? What was the number of seizures before diagnosis? What countries are the patients from?
h) Since when the following up took place? What was the ethnicity of the patients? Were there neuroimaging abnormalities? What were the EEG and video-EEG abnormalities? Did they have febrile seizures, or neonatal seizures, what was their gestational age? Did they have a low weight at birth? The developmental delay should be further characterized as well as neurological examination. What was their seizure onset type? Do they have drug-resistant epilepsy? All of these characteristics and many more have been associated with prognostication in drug-resistant epilepsy and should not be ignored.
7) Was this study protocol based on previous studies? Why were these tests performed? What were the researchers hoping to achieve with these tests? What have they done differently from other studies? How does their research finding differ from previously published literature? What does this research add of new? How does this research contribute to clinical practice, and what are the practical implications and the clinical relevance of this study?
8) Were there controls in all of the experiments?
9) The authors should review reporting guidelines for experimental studies and adequate their introduction, abstract, methods, results, and discussion sections.
10) In general, authors should review the text and add all the relevant citations through the text as many seem to be missing. Also, authors should work with English editing services to improve readability of the article.

Author Response
We acknowledge the general comments of the reviewer. The general guidelines of their comments have been considered, although some clinical details requested by the referee are not available to us. Indeed, this article is not focused on the clinical aspects (many patients with mutation in SLC6A1 gene have been previously reported), but in the molecular mechanisms of the dysfunctions. Much less is known about this. Nevertheless, we will try to address the suggestions:
1) It would be interesting if the authors could include the type of research design on the title.
We have modified the title to include the type of research design. The new title is “Experimental and bioinformatic insights into the effect of epileptogenic variants on the function and trafficking of the GABA transporter GAT-1.”
2) The abstract should mention the research goals.
We have modified the abstract to state the research goals.
3) L37 : I would advise removing “reviewed in” and only citing the reference.
We have deleted “reviewed in.”
4) L82-93: The authors should clearly state in this paragraph what is the research question? What was the main goal of the research? What were the secondary goals? And why is this research relevant? Also, it seems that along these lines the authors are describing the findings of the research. If this is the case, it is better than the authors rewrite this paragraph and add the findings of the research to the results/discussion section. However, if those are findings from other studies, appropriate references should be added.
The final paragraph of the introduction has been re-written, including the point raised by the reviewer (lines 83-95)
5) L130 : Do the authors mean the absence of Landau reflex?
Yes, this has been amended.
6) L546: In the materials and methods section, the authors should explain and further describe:
The clinical information relative to the two patients has been distributed between the Results and Method sections, trying to include the point raised by the reviewer.
- a) Study design
The findings of the molecular disfunction of these patients is a consequence of the clinical procedures of the two public hospitals involved in this research (now in Methods).
- b) The authors should describe in detail which statistical analyzes were performed. They should also report confidence intervals and p-values, as well as which parameters were analyzed and how. They should also describe the computer program used to carry out the analyzes.
We have included a section on statistical methods in the Materials and Methods. These are not applicable to the clinical part since only two patients were available. (p22)
- c) Timeframe of the study
The timeframe before diagnosis is indicated in the Results section.
- d) Setting of the study. Was it conducted in a public hospital? Private hospital? Clinic?
Patients were identified in public hospitals. Research was carried out in public research institutes. This information is now included.
- e) A table with baseline characteristics of the two cases should be provided.
Table 1 now includes a summary of these data.
- f) The authors should provide further clinical and epidemiological information about the patients.
We have included the available clinical data in the Results section.
- g) Which type of patient population is this? Which is the age of epilepsy diagnosis? Which drugs are they on? Were other AED tried? What is the frequency of the seizures per day? Do they have more than one type of seizure? How many drugs were tried? Did they have status epilepticus? How many years of following-up do these patients have? What was the number of seizures before diagnosis? What countries are the patients from?
These data, when available, are included in the Results section. The complete 2.1. section has been rewritten.
- h) Since when the following up took place? What was the ethnicity of the patients? Were there neuroimaging abnormalities? What were the EEG and video-EEG abnormalities? Did they have febrile seizures, or neonatal seizures, what was their gestational age? Did they have a low weight at birth? The developmental delay should be further characterized as well as neurological examination. What was their seizure onset type? Do they have drug-resistant epilepsy? All of these characteristics and many more have been associated with prognostication in drug-resistant epilepsy and should not be ignored.
These data, when available, are included in the Results section 2.1.
7) Was this study protocol based on previous studies? Why were these tests performed? What were the researchers hoping to achieve with these tests? What have they done differently from other studies? How does their research finding differ from previously published literature? What does this research add of new? How does this research contribute to clinical practice, and what are the practical implications and the clinical relevance of this study?
A new paragraph has been added to the Discussion to better define the points raised by the reviewer (l. 614-620)
8) Were there controls in all of the experiments?
We think all the experiments have adequate positive and negative controls.
9) The authors should review reporting guidelines for experimental studies and adequate their introduction, abstract, methods, results, and discussion sections.
Guidelines of the journal have been considered. New paragraphs have been included in the Introduction, Results and Discussion.
10) In general, authors should review the text and add all the relevant citations through the text as many seem to be missing. Also, authors should work with English editing services to improve readability of the article.
Ten additional references have been included (numbers 3,18,23,24,28,34,38,40,42,59). The text has been reviewed by a professional scientific editor from the USA, and the updated version has also been reviewed by this same editor.
Round 2
Reviewer 1 Report
All my concerns have been addressed. The article is now suitable for publication.